# Label-Free Detection of Zeptomol miRNA via Peptide Nucleic Acid Hybridization Using Novel Cyclic Voltammetry Method

**DOI:** 10.3390/s20030836

**Published:** 2020-02-04

**Authors:** Shintaro Takase, Kouta Miyagawa, Hisafumi Ikeda

**Affiliations:** 1New Business Management Division, Management Planning H.Q., YOKOWO CO. LTD., 5-11, Takinogawa 7-Chome, Kita-ku, Tokyo 114-8515, Japan; k-miyagawa@jp.yokowo.com; 2Department of Environmental Science and Education, Faculty of Home Economics, Tokyo Kasei University, 1-18-1 Kaga, Itabashi City, Tokyo 173-8602, Japan; ikeda-h@tokyo-kasei.ac.jp

**Keywords:** new ion-channel sensor model, cyclic voltammograms, *ΔE* measurements, PCR-less, label-free, new hybridization system, 140-zeptomol sensitivity

## Abstract

To harness the applicability of microribonucleic acid (miRNA) as a cancer biomarker, the detection sensitivity of serum miRNA needs to be improved. This study evaluated the detection sensitivity of miRNA hybridization using cyclic voltammograms (CVs) and microelectrode array chips modified with peptide nucleic acid (PNA) probes and 6-hydroxy-1-hexanethiol. We investigated the PNA probe modification pattern on array chips using fluorescently labeled cDNA. The pattern was not uniformly spread over the working electrode (WE) and had a one-dimensional swirl-like pattern. Accordingly, we established a new ion-channel sensor model wherein the WE is negatively biased through the conductive π–π stacks of the PNA/DNA duplexes. This paper discusses the mechanism underlying the voltage shift in the CV curves based on the electric double-layer capacitance. Additionally, the novel hybridization evaluation parameter *ΔE* is introduced. Compared to conventional evaluation using oxidation current changes, *ΔE* was more sensitive. Using *ΔE* and a new hybridization system for ultrasmall amounts of aqueous solutions (as low as 35 pL), 140 zeptomol label-free miRNA were detected without polymerase chain reaction (PCR) amplification at an adequate sensitivity. Herein, the differences in the target molar amount and molar concentration are elucidated from the viewpoint of hybridization sensitivity.

## 1. Introduction

Since the initial discovery of miRNA [1], its functions have been investigated worldwide, particularly with regard to cancer. In fact, miRNA contributes to tumorigenesis through numerous genetic mechanisms, and its expression profiles in serum or plasma serve as potential noninvasive diagnostic and prognostic biomarkers for cancer [2,3,4,5,6,7,8].

To harness the applicability of miRNA as a cancer biomarker, we must establish appropriate methods for its detection. Among the many requirements that must be satisfied to accomplish this, a high miRNA detection sensitivity is paramount because miRNA levels in the human blood are very low. The levels of circulating miRNA in cancer patients depend on the miRNA type, of which mir-21-5p is present at relatively high levels. However, its mean serum levels are approximately 50–100 copies/μL [9], corresponding to 83–166 zmol/mL. Even if 10 mL of serum is available during cancer screening, miRNA levels will only be approximately 830 zmol to 1.7 amol.

Currently, the fluorescent labeling of target nucleic acids is a gene detection method that is widely used in genetic diagnoses, facilitating the simultaneous detection of thousands of genes [10,11,12,13,14]. However, these methods are time-consuming, costly [15], and a potential cause for concern if the fluorescent labeling alters the hybridization. Alternatively, label-free electrochemical detection has many advantages, including simplicity, rapidity, low cost, and ease of analysis. Aoki et al. [15] developed electrochemical gene sensor array chips based on the principle of ion-channel sensing with peptide nucleic acid (PNA) probes on ∅1.6-mm gold electrodes and detected 100 μM of target DNA. By estimating an aliquot amount of 10 μL or more for hybridization from the modification data of 6-hydroxy-1-hexanthiol (6-HHT), this method yields a target DNA level of approximately 1 nmol or more, which greatly exceeds the required miRNA detection sensitivity of approximately 1.7 amol or less. The further improvement of label-free electrochemical gene detection sensitivity is highly anticipated.

Peterson et al. [16] investigated the influence of DNA probe density on the gold substrate surface of a DNA target using surface plasmon resonance. They changed the probe density from 2.0 × 10^12^ to 12.0 × 10^12^ molecules/cm^2^ and demonstrated that the hybridization efficiency strongly depended on the probe density. In other words, the efficiency increased as the density decreased. Because DNA has negative charges, DNA probe modification should be uniform due to the electrostatic repulsion between DNA probes. Therefore, the DNA probe average density is a good parameter for evaluating hybridization efficiency. Modified probe uniformity over an array chip surface is also implicit in an ion-channel sensing model [17,18,19]. However, this may not be the case for PNA probes, which do not possess negative charges [20].

The objective of this study was to evaluate the detection sensitivity of miRNA hybridization using PNA probes in cyclic voltammograms (CVs). We investigated the PNA probe modification pattern on an array chip surface using fluorescently labeled cDNA and observed a one-dimensional swirl-like pattern. Accordingly, we established a new ion-channel sensor model and introduced a novel evaluation parameter for hybridization, namely, *ΔE*, into the cyclic voltammetry measurements. Compared to conventional evaluation parameters, *ΔE* has a higher sensitivity. This model also predicts that the sensitivity of *ΔE* improves if the size of the working electrode (WE) is reduced. Furthermore, we developed a new hybridization system that minimizes the amount of the aqueous solution and target miRNA involved and then confirmed the detection sensitivity of 140 zmol using a ∅67-μm WE.

## 2. Materials and Methods

### 2.1. Reagents

Antiparallel PNA probes were synthesized using Panagene (Daejeon, South Korea) and modified using H-(Cys-(AEEA)). Here, Cys refers to cysteine, which is used for covalent attachment to gold electrodes using gold/thiol bond formation [15,16,21] and AEEA refers to 2-aminoethoxy-2- ethoxyacetic acid. Their sequences were as follows:PNA(N1); H-(Cys-(AEEA)-TCGATGTAACAGACGACC)-NH_2_,PNA(N3); H-(Cys-(AEEA)-ATGTCATGACACTATTGACTT)-NH_2_, andPNA(N16); H-(Cys-(AEEA)-ATCGTCGTGCATTTATAACCGC)-NH_2_.

The target miRNA and DNA were purchased from Eurofins Genomics (Tokyo, Japan). Their sequences were as follows:miRNA(R3); 5’-r(UACAGUACUGUGAUAACUGAA)-3’ andDNA(D1); 5’-d(AGCTACATTGTCTGCTGG)-3’.

The miRNA (R3) sequence complemented that of the PNA probe (N3), whereas the DNA (D1) sequence complemented that of another PNA probe (N1).

Analytical reagent-grade chemicals were used along with Milli-Q reagent-grade water (Millipore, Bedford, MA, USA).

### 2.2. Microelectrode Array Chip

A 120-channel gold microelectrode array chip was particularly designed for statistical data analysis and to perform various PNA probe modifications (Kyodo International, Inc., Kawasaki, Japan). We designed the 120-channel microelectrode array chip with reference to the basic structure of sensor array chips reported in a previous study [15], with a few modifications in the materials used and the dimensions. The array chip was fabricated on a glass substrate of 25.8 × 25.8 mm^2^ (BK7) and patterned by a chromium adhesion layer (500 Å) and a sputtering gold layer (2000 Å). To achieve high sensitivity, the diameters of the working electrodes (WEs) were limited to ∅67 μm and ∅95 μm through openings in a photosensitive epoxy resin (S2000) that was spin-coated over the metal layers and glass substrate. Figure 1a depicts the scanning electron microscope (SEM) image of one block of the microelectrode array chip, which contained 15 pieces (pcs) of WEs (top view, ∅67 μm). Figure 1b presents the SEM image of a single WE (45° tilted view, ∅67 μm).

### 2.3. Array Chip Preparation

We developed the process of preparing the array chip by referring to the processes and conditions of the aforementioned study [15]. The microelectrode array chips were surface-cleaned under oxygen plasma using a plasma ion bombarder (PIB-10, Vacuum Device; Ibaraki, Japan) at 20 Pa for 2 min before modification of the gold electrodes with the PNAs and 6-HHT. Next, the 120-channel gold electrodes were divided into eight blocks (15 channels per block) using a punched parafilm placed on the array chip for the various PNA modifications. Ten-microliter aliquots of 10-μM aqueous PNA probes were poured into each block, and the PNAs were modified at 25 °C for 30 min. The PNAs were modified by placing the array chip and a wet gauze inside a small plastic case equipped with a lid to prevent the evaporation of aliquots. After being rinsed with Milli-Q water at 25 °C, the array chip was immersed in 100 mL of Milli-Q water at 80 °C for better cleaning. Thereafter, the 200-μL aliquots of the 1-mM aqueous 6-HHT solution were poured onto the surface of the entire array chip at 25 °C for 30 min. After being rinsed with Milli-Q water at 25 °C, the array chip was subjected to electrochemical measurements.

### 2.4. Electrochemical Measurements

We developed a 120-channel potentiostat that could perform cyclic voltammetry (CV) and square wave voltammetry measurements. Sixteen channels were simultaneously used for assessment, and the assessment was repeated eight times to acquire data from all electrodes on the array chip: the measured data were stored on a personal computer. During the CV measurements, the applied voltage was set to ±2 V at 62.5-μV increments, with a maximum voltage scan speed of 1 V/s. The current measurement range was selected from two ranges: ±200 nA (max) and ±2 μA (max) at a resolution of 25 fA. The background current was suppressed to 0.5 pA (rms) by setting the current–voltage converter close to the array chip. 

Using this potentiostat, electrochemical measurements were carried out using an RE-1B reference electrode (RE; BAS Inc. Tokyo, Japan) and a platinum counterelectrode (CE; BAS Inc.) in an aqueous solution of 0.25 mM of phosphate buffer and 0.5 mM of NaClO_4_ with 1 mM [Fe (CN)_6_]^4−^ as an electroactive marker. The applied voltage range was −0.15 to +0.5 V at a scan speed of 500 mV/s.

## 3. Results and Discussion

### 3.1. New Ion-Channel Sensing Working Model

With regard to electrochemical gene sensor array chips based on ion-channel sensing, Aoki et al. [17,18] have discussed the working principle of a sensor with oligonucleotide targets and PNA probes. Before PNA/DNA hybridization, the negatively charged markers can access the electrode and transfer electrons to the WE. After hybridization, the electrostatic repulsion between the negatively charged marker and the PNA/DNA duplexes on the WE surface hinders the redox reaction in the marker. We term the abovementioned mechanism an ordinary sensor working model: this mechanism is depicted in Figure 2a. This working model was established based on the assumption that PNA/DNA duplexes are uniformly spread over the WE. 

We assessed the patterns of the hybridized PNA/DNA duplexes over the WE modified by antiparallel PNA (N1) and 6-HHT using fluorescently labeled cDNA (D1, carboxytetramethylrhodamine (TAMRA)). The electrode diameter was ∅300 μm, and the target cDNA concentration was 100 nM. 

Figure 2b shows the fluorescence hybridization images. The PNA/DNA duplexes were not uniformly spread over the WE but rather exhibited a one-dimensional swirl-like pattern. This fluorescence pattern was considered to be a modified PNA (N1) pattern in which the fluorescently labeled cDNA (D1) was hybridized. The self-assembled 6-HHT monolayer had high orientation and stability [21]. Therefore, the 6-HHT modification after PNA modification may have helped align the PNAs in a swirl-like pattern because PNA is electrically neutral. A large area of the WE did not contain the PNA/DNA duplex. Therefore, the previously mentioned working principle does not explain the mechanism governing the hindrance of the redox reaction over the entire WE.

Eley and Spivey [22] have reported on the conductivity of dsDNA and proposed a conductivity model for transferring π electrons through the π–π stacks of the base pairs. After this, numerous controversial studies have investigated whether dsDNA is conductive [23,24,25,26,27,28,29]. Guo et al. [30] assessed dsDNA conductivity by attaching carbon nanotubes to dsDNA and reported that dsDNA without mismatches was as conductive as graphite, while the mismatched dsDNA had 300-fold greater resistivity. Therefore, the matched PNA/DNA and PNA/miRNA duplexes should be conductive through their π–π stacks. 

The new sensor working model is presented in Figure 2c. Prior to PNA/DNA hybridization, negatively charged markers can access the electrode and transfer electrons to the WE, similarly to the original sensor working model. However, the modified PNA pattern after 6-HHT modification is a one-dimensional swirl-like pattern and is not uniformly spread over the WE. After hybridization, the negative charge in the backbone of the hybridized oligonucleotides is electrically connected to the WE through the π–π stacks of the hybridized base pairs. As the surface of the WE is a gold layer, this electrically connected negative charge generates a uniform negative potential over the entire WE and hinders the redox reaction of the marker.

### 3.2. ΔE Potential Shift

The total negative charge of the hybridized oligonucleotides, *ΔQ*, leads to a potential difference *ΔE* in the electric double layer capacitance *C* when measuring the CV as follows:(1)ΔE=ΔQ/C.

The diagrams show the differences in the electronic potential between the reference electrode (RE) and the WE pre- and posthybridization. During prehybridization, the electronic potential between the RE and WE that is required to yield the current (*im*) is shown in Figure 3a, where the external bias can be expressed as follows:(2)$$V_{0}=\left(\Delta \varphi_{0r}+\Delta \varphi_{b}\right)+\left(\Delta V_{b0}+\Delta \varphi_{0w}\right).$$

After hybridization, the electronic potential between the RE and WE decreases by *ΔE* due to the negative charge on the hybridized oligonucleotide, *ΔQ*, as shown in Figure 3b. There will be a reduction in the current if the external bias is identical to *V_0_*. To retrieve the identical current (*im*), the external bias *V_1_* must be increased by *ΔE* to yield the same potential difference between the RE and WE, as shown in Figure 3c and expressed by Equation (3):(3)$$V_{1}=V_{0}+\Delta E,$$
where *V_0_:* the external bias to yield (*im*) in prehybridization; *V_1_:* the external bias to retrieve (*im*) in posthybridization; *ΔE:* the potential difference caused by the hybridized charge *(ΔQ)*; *ΔΦ_0r_:* the potential difference between the RE and the aqueous solution (spontaneous potential); *ΔΦ_b_*: the potential difference in the bulk aqueous solution (generally close to zero); *ΔΦ_0w_*: the potential difference between the WE and the aqueous solution (spontaneous potential); and *ΔV_b0_*: the potential difference controlled by external bias (*V_0_*).

Similar equations have been proposed with regard to DNA target detection sensitivity using ion-sensitive field effect transistors (IS-FETs) [31] and potentiometry [32]. Ohtake [31] investigated the effect of using DNA or PNA probes with IS-FET electrodes on target DNA hybridization. He calculated the increase in the quantity of electricity per gate area using Equation (4), where *ΔV_T_* is the threshold voltage shift of the IS-FET caused by hybridization and *L* and *W* are the gate length and width, respectively. Here, *C* is the capacitance of the gate fabricated using SiO_2_ (100 nm), Si_3_N_4_ (100 nm), and Ta_2_O_5_ (40 nm) thin films and not the electric double-layer capacitance, as expressed in Equation (1):(4)$$\Delta Q=\left(C\Delta V_{T}\right)/\left(\mathrm{LW}\right).$$

Goda et al. [32] have investigated the sensitivity of potentiometric DNA detection performed using PNA and DNA probes. Using Equation (5), they discussed the potential shift caused by hybridization. Here, *C_DL_* is the electric double-layer capacitance, as in Equation (1). However, *ΔQ* is the negative charge of the captured DNA within the electrical double layer. Goda et al. insisted that the DNA located outside of the electrical double layer fails to generate an interfacial potential as a result of the charge-screening effect caused by the external electrolytes in the buffer solution: (5)$$\Delta V=\left(Q+\Delta Q\right)/\left(C_{DL}+\Delta C_{DL}\right)-\left(Q/C_{DL}\right).$$

Apart from the differences between the equations used in this study and Equations (4) and (5) in terms of parameter definition, our new model is different from existing models because our objective is to understand the complete mechanism governing the influence of the negative charge of the captured DNA on the potential shift of gold WEs through electric double-layer capacitance. Therefore, discovering the one-dimensional swirl-like pattern of PNAs and the mechanism of the electrical conductivity caused by the π electrons in the PNA/DNA duplexes is important. For example, Goda et al. [32] discussed the average surface density of immobilized PNAs; however, the PNA pattern was not considered. Additionally, their discussion with regard to Equation (5) was based on an analogy of the IS-FET model. However, as long as the electric double-layer capacitance is used instead of the gate capacitance, *ΔQ* should exist on the gold electrode (on which the PNA or DNA probes are modified) and will not be confined to the electrical double layer itself, as is known from basic electromagnetic theory.

### 3.3. ΔE Evaluation

The potential shift *ΔE* was investigated and compared to the conventional evaluation parameter [15] of hybridization in the CV measurements, that is, the changes in the oxidation current. Five blocks were modified using the PNA probe (N3) and 6-HHT. After the prehybridization CV measurements, miRNA (R3) hybridization was performed at 40 °C for 40 min with an aqueous solution of 1 × saline sodium citrate (SSC) and 20% dimethyl sulfoxide (DMSO). Thereafter, we performed posthybridization CV measurements. 

Figure 4a compares the curves between the pre- and posthybridization CVs. The prehybridization curve is indicated by the solid line, and the posthybridization curves are indicated by the dotted lines, where the short dots denote 240 nM and the long dots denote 4 nM. In the conventional evaluation method, the sensor response is expressed as the ratio of posthybridization current reduction (*i_0_* – *i*) to the prehybridization peak current (*i_0_*), where (*i*) is measured at potential (*E_p_*), determining the peak current (*i_0_*). When the target miRNA levels were markedly high, that is, up to 240 nM, the posthybridization CV current at *Ep* was suppressed to a low value (*i’*) in accordance with the hybridized negative charge of the target miRNA. Therefore, the conventional evaluation parameter, that is, *(i_0_ - i)/i_0_* or *i/i_0_*, can be useful in evaluating the degree of hybridization.

However, evaluations based on this parameter have several disadvantages. First, in the prehybridization CV measurements, we could not explicitly set the peak current voltage *Ep* due to gradual changes in the oxidation current around the peak current. When the oxidation current approached its peak value, there was a decrease in the concentration of the reductant marker [Fe(CN)_6_]^4–^, whereas the concentration of the oxidant [Fe(CN)_6_]^3–^ increased. At peak current, the concentration gradient of the reductant perpendicular to that of the WE peaked, which yielded the peak current value. Along with this gradual change in the gradient, the oxidation current also gradually changed, which resulted in errors when setting the *Ep* value and evaluating *(i_0_ - i)/i_0_* or *i/i_0_*. 

Another difficulty occurred when the amount of the hybridized target was as low as 4 nM. The current *i*, at the *Ep* on the posthybridization CV, was approximately identical to that of the *i_0_* on the prehybridization CV. Thus, the evaluation parameter *i/i_0_* remained constant at approximately one.

Compared to the prehybridization CV, the posthybridization CV exhibited a shift to a higher potential value, as expressed by Equation (3), and maintained a similar oxidation curve. Therefore, the voltage shift value *ΔE* at an identical measurement current (*im*) is a potentially good evaluation parameter for the amount of hybridized target miRNA, because it is based on the simple physical model of Equation (1). 

Figure 4b shows a 74-pc correlation chart of *i/i_0_* and *ΔE* when the concentration of the complementary target miRNA (R3) was 4 nM. Because the target miRNA concentration was low, the majority of the *i/i_0_* plots tending toward one remained unchanged and exhibited poor detection sensitivity for the hybridized target miRNA. However, the *ΔE* plots had a wide spread from 0 to 90 mV. Compared to the conventional *i/i_0_* evaluation parameter, *ΔE* had higher sensitivity and was better suited to the detection of hybridized miRNA at a low concentration.

### 3.4. New Hybridization System for Ultrasmall Aqueous Solution Amounts

To detect the small amount of miRNA (1.7 amol or less) in the serum through *ΔE* measurements, the volume of the aqueous solution for hybridization must be limited to 425 pL or less to maintain the concentration at 4 nM. Figure 5 shows a comparison between (a) an ordinary hybridization system and (b) a new hybridization system, where a small well caused by an opening in the epoxy resin and Teflon plate limits the volume of the aqueous solution. This sandwich structure ensures a constant small well volume over each WE and allows for comparing the hybridization results between the WEs. Moreover, the Teflon plate suppresses the evaporation of the aqueous solution during hybridization and prevents the aqueous solution from being removed from each well through capillary forces due to its hydrophobicity property when pressed to the array chip by a metal weight.

In the new hybridization system, the size of the WE must be considered from the point of view of sensitivity. Assuming two differently sized WEs, e.g., WE_1_ and WE_2_, are used to detect the same minimal potential shift *ΔE* in Equation (1), the following Equation (6) can be derived: (6)$$\frac{\Delta Q_{1}}{\Delta Q_{2}}=\frac{C_{1}}{C_{2}},$$
where *ΔQ_1_:* the charge required to achieve the minimal potential shift *ΔE* in *WE_1_*; *ΔQ_2_*: the charge required to achieve the minimal potential shift *ΔE* in *WE_2_*; *C_1_*: the electric double-layer capacitance of *WE_1_*; and *C_2_*: the electric double-layer capacitance of *WE_2_*.

If the area of WE_1_ is less than that of WE_2_, the electric double-layer capacitance *C_1_* is less than *C_2_,* because the capacitance is proportional to the area of the WE. Therefore, the charge required to achieve the same minimal potential shift *ΔE* becomes smaller in *ΔQ_1_* compared to *ΔQ_2_*. To confirm this, two well-sized array chips were prepared in the new hybridization system by changing the well sizes to ∅67 μm/t10 μm and ∅95 μm/t10 μm: the respective well volumes were 35 pL and 71 pL. Both of the well volumes were significantly smaller than the target well volume (less than 425 pL). Here, ∅67 μm was selected, considering the ease of processing the photosensitive epoxy resin to obtain a clean WE surface without etching debris, and ∅95 μm was selected to make the area of the WE approximately twice of ∅67 μm (to conform to Equation (6)). 

### 3.5. MiRNA Detection Sensitivity

Five of the eight blocks in the array chip were modified using a PNA probe (N3) for complementary miRNA hybridization. Two more blocks were modified by PNA (N1) and (N16) for noncomplementary hybridization. The remaining block was not modified by PNA to check whether there were any target miRNA chemisorbs and remained on the 6-HHT-modified working electrodes. The *ΔE* detection sensitivity data are box-plotted in Figure 6a (35 pL (∅67 μm)) and Figure 6b (71 pL (∅95 μm)) at various target miRNA molar amounts. In each box plot, (N3/R3) indicates 75 pcs of data of a complementary system, (N1/R3) and (N16/R3) each refer to 15 pcs of data of noncomplementary systems, and (PNA-less/R3) indicates 15 pcs of data of a reference system (to identify the occurrence of the chemisorption of miRNA onto 6-HHT). 

In Figure 7a, the median values of the complementary system (N3/R3) in Figure 6a,b are plotted on a linear scale as a function of the molar amount *N*. The smaller well volume of 35 pL had the highest *ΔE* detection sensitivity at 140 zmol. The sensitivity increased with the decrease in the WE size, as predicted in Section 3.4. When *ΔE* was replotted against the target molar concentration *D*, the sensitivity to the well volume varied, as shown in Figure 7b. The sensitivity was the highest at 4 nM for the well volume of 35 pL (∅67 μm). In terms of molar concentration, the sensitivity appeared to depend less on the size of the WE, while the sensitivity appeared to depend more on the size of the WE in terms of the molar amount.

Equation (1) can be modified to Equation (7), where *N* is the target miRNA molar amount, and *α* (0 ≤ *α* ≤ 1) is the hybridization efficiency representing the hybridized miRNA molar amount divided by the total miRNA molar amount originally preserved inside the well:(7)$$\Delta E=\frac{qmN_{A}d_{0}}{\varepsilon \varepsilon_{0}}\left(\frac{\alpha }{S}\right)N,$$
where *q:* the charge of the electrons; *m:* the base length of target miRNA; *N_A_*: Avogadro’s number; *d_0_:* the thickness of the electric double-layer capacitance; *ε:* the permittivity of the electric double-layer capacitance; *ε_0_:* the permittivity of the vacuum; *α:* the hybridization efficiency (0≤ *α* ≤1); *S:* the area of the WE; and *N:* the target miRNA molar amount in the well.

To achieve a higher detection sensitivity in the molar amount, the gradient of Equation (7) should be increased. Therefore, decreasing *S*, that is, the area of the WE, and increasing the hybridization efficiency, *α*, are both effective. The nonlinear curve in Figure 7a means that *α* has nonlinear characteristics. This may be the reason for the tangential line around the lowest molar amount not crossing the origin. In Figure 7a, the gradient of Equation (7) at the lowest molar amount is calculated as 5.86 × 10^16^ (V/mol) for ∅67 μm and 8.58 × 10^15^ (V/mol) for ∅95 μm. By taking the ratio of these gradients, the hybridization efficiency ratio of the two differently sized WE can be calculated, as expressed in Equation (8). As expected, decreasing *S* by decreasing the diameter from ∅95 μm to ∅67 μm contributes to an increase in sensitivity. However, the ratio of *S* is two, whereas the ratio of *α* is 3.4 and 1.7 times higher than the ratio of *S*. This means that the hybridization efficiency *α* is strongly dependent on the WE area:(8)$$\frac{\alpha_{\emptyset 67}}{\alpha_{\emptyset 95}}=3.4.$$

Equation (7) can be modified to Equation (9) using the molar concentration *D,* where *t* is the thickness of the well. Notably, Equation (9) expresses that decreasing the WE size does not directly increase the molar concentration sensitivity; rather, only the hybridization efficiency *α* does:(9)$$\Delta E=\frac{qmN_{A}d_{0}t}{\varepsilon \varepsilon_{0}}\alpha D,$$ where *t* is the thickness of the well, and *D* is the target miRNA molar concentration in the well.

The increase in the hybridization efficiency *α* due to the decreasing WE size is probably caused by the differences in the PNA modification pattern. The one-dimensional swirl-like PNA pattern may be more congested in WEs of a smaller size. This is plausible because the congestion degree of the PNA pattern directly dictates the interaction probability between the PNA probe and the target miRNA. In other words, as the congestion degree becomes higher, the interaction probability increases.

Unfortunately, the congested one-dimensional PNA pattern is not as effective for hybridization as the PNA probes uniformly spread over the WEs. However, the realization of the latter remains unclear. Using DNA probes may be an effective solution; however, this decreases the hybridization with negatively charged miRNA. If the PNA modification pattern can be changed from one-dimensional to two-dimensional while keeping a reasonable distance between the PNA probes, a remarkable sensitivity improvement will occur in terms of molar concentration.

### 3.6. Statistical Analysis

We statistically analyzed the *ΔE* data obtained from all tests. To specify the highest *ΔE* detection sensitivity of this incubation system from a statistical perspective, we performed an unpaired test (two-tailed) between the 75 pcs of complementary data (N3/R3) and the other systems (N1/R3, 15 pcs; N16/R3, 15 pcs; PNA-less/R3, 15 pcs), followed by determining the *p*-value. To prevent a type 1 error in this multiple comparison, we performed Dunnett’s test. Equation (10) yields the *t*-value using the *V_E_* expressed by the unbiased variances in Equation (11), as follows:(10)$$t_{1i}=\frac{{\bar{X}}_{1}-{\bar{X}}_{i}}{\sqrt{V_{E}\left(\frac{1}{N_{1}}+\frac{1}{N_{i}}\right)}},$$
(11)$$V_{E}=\frac{\sum_{i=1}^{a}\left(N_{i}-1\right)u_{i}^{2}}{\left(N_{1}+N_{2}\cdots N_{a}-a\right)}.$$

The *p*-value (based on Dunnett’s test) only indicates a rare risk where there are no differences between the mean values. To determine the degree of separation between the mean values, the effect size was evaluated using Cohen’s *d*-value (defined in Equation (12)), where the unbiased variances in Equation (13) express *u_d_* as follows:(12)$$d=\frac{\bar{X}-\bar{Y}}{u_{d}},$$
(13)$$u_{d}=\sqrt{\frac{\left(N_{1}-1\right)u_{X}^{2}+\left(N_{2}-1\right)u_{Y}^{2}}{N_{1}+N_{2}-2}}.$$

Figure 8a,b shows the calculated Dunnett’s *p*-value and Cohen’s *d*-value. Three groups were used for statistical analysis: (N3/R3) versus (N1/R3), (N3/R3) versus (N16/R3), and (N3/R3) versus (PNA-less/R3) for the 35-pL and 71-pL hybridization.

Dunnett’s test revealed that in the 35-pL hybridization system, if the miRNA levels were 140 zmol or greater, the mean value of the complementary and noncomplementary systems was not equal at a 95% significance level. Moreover, Cohen’s *d*-value revealed that, under this condition, the mean value of the complementary and noncomplementary systems had good separation that exceeded the medium (≥0.5) or large (≥0.8) effect size criteria. Figure 8 shows that the possibility of miRNA chemisorption to 6-HHT was low.

These statistical analyses indicate that in the 35-pL system, the maximum *ΔE* detection sensitivity was 140 zmol. This new hybridization methodology exhibited good sensitivity against our initial target value of 830 zmol to 1.7 amol. Compared to the calculated sensitivity of 1 nmol (discussed in Section 1), we achieved a 10^10^-fold improvement for the molar sensitivity, which implies that in an ordinary incubation system, most miRNA targets are wasted, while only the targets close to the WE contribute to the hybridization results.

### 3.7. Two Aspects of Sensitivity

Sensitivity in a hybridization system has two aspects. One is the measurement sensitivity itself, which is associated with the molar amount of the target. The ultrasmall amount of the incubation system in the aqueous solution revealed that the novel *ΔE* measurement and smaller WE could detect miRNA amounts as low as 140 zmol with a ∅67-μm WE with adequate sensitivity. 

The second aspect is associated with the target molar concentration. Figure 7b shows that the highest sensitivity for a molar concentration with a ∅67-μm WE was 4 nM when the incubation condition was 40 °C for 40 min. For successful hybridization, there should initially be a reasonable interaction probability between the PNA and the miRNA. Thereafter, hydrogen bonding occurs between the bases, followed by π–π stacking to complete the hybridization process. A reasonably high molar concentration of miRNA ensures the occurrence of this interaction. Even if 1.7 amol of miRNA were present in the 10 mL of serum, the molar concentration was 0.17 fM, which is much lower than 4 nM and does not yield a reasonable interaction probability. 

To accomplish the direct detection of miRNA in serum, these two aspects of sensitivity must be fulfilled simultaneously. From this viewpoint, increasing the sensitivity in terms of the molar concentration becomes an issue. One approach is to investigate a method of changing the PNA modification pattern from one-dimensional to two-dimensional to dramatically increase the hybridization efficiency α. The other approach is to extract and concentrate the miRNA from the serum and deliver it to a small incubation well for hybridization. 

## 4. Conclusions

This study investigated an ion-channel sensing model. On the basis of the PNA/DNA duplex fluorescence pattern, which was a one-dimensional swirl-like pattern nonuniformly spread over the WE, we proposed a new model wherein the negative charges of the PNA/DNA (and PNA/miRNA) duplexes develop a uniform negative potential over the WE through their conductive π–π stacks, which hinder the redox reactions of the marker. Within the context of this model, the novel evaluation parameter *ΔE* was introduced and defined as the hybridized negative charge of oligonucleotides, *ΔQ,* divided by the electric double-layer capacitance *C*. This model is different from the IS-FET- and potentiometry-based models discussed in References [31] and [32], respectively. Moreover, the *ΔE* sensitivity was investigated through the development of a new hybridization system that uses an ultrasmall amount of aqueous solution, which can be as low as 35 pL. The maximum detection sensitivity, which was 140 zmol at a 95% significance level, was achieved for the smaller WE (∅67 μm), as was expected based on the model. This system had sufficiently high molar amount sensitivity for the detection of cancer-related miRNA in 10 mL of serum without PCR amplification. 

The sensitivity of a hybridization system has two aspects, namely, the target molar amount and the target molar concentration, where both must have sufficient sensitivity at the same time for the direct detection of miRNA in serum. To this end, increasing the sensitivity of *ΔE* in terms of molar concentration, which is currently 4 nM and markedly below the required 0.17 fM, is an issue that should be addressed in future work. The congestion degree of the one-dimensional PNA modification pattern probably determines the interaction probability between the PNA probe and the target miRNA. Therefore, it is important to investigate how the pattern of PNAs can be changed from one-dimensional to two-dimensional to improve the hybridization efficiency *α* and/or to examine how to extract and concentrate miRNA from serum.

## Figures and Tables

**Figure 1 sensors-20-00836-f001:**
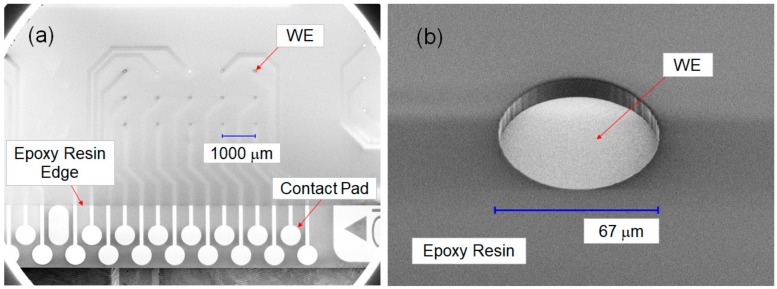
(**a**) SEM image of one block of the microelectrode array chip, which contained 15 pieces (pcs) of working electrodes (WEs) (top view, ∅67 μm). (**b**) SEM image of a single WE (45° tilted view, ∅67 μm).

**Figure 2 sensors-20-00836-f002:**
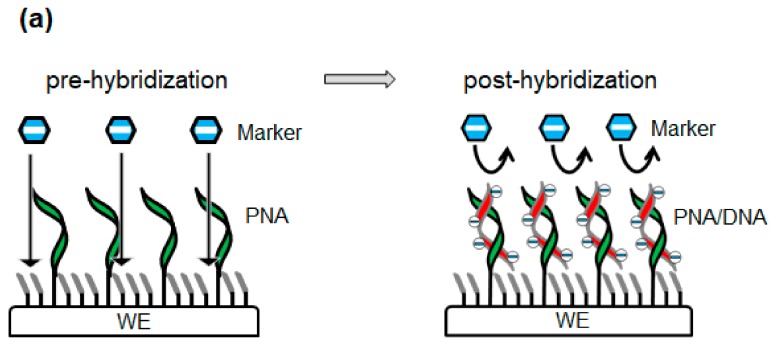
(**a**) Ordinary sensor working model where the PNA is uniformly distributed over the WE. Prior to hybridization, negatively charged markers can access and transfer electrons to the WE. After hybridization, electrostatic repulsion between the negatively charged marker and the uniformly distributed PNA/DNA duplexes at the WE surface hinders the redox reaction of the marker. (**b**) Fluorescence image (OLYMPUS Fluorescent Microscope; BX53M) of peptic nucleic acid (PNA)(N1)/DNA(D1) duplexes obtained after the fluorescent labeling of cDNA (carboxytetramethylrhodamine (TAMRA)). The electrode diameter was ∅300 μm, and the target DNA concentration was 100 nM. The fluorescence pattern was not uniformly spread over the WE, but rather exhibited a one-dimensional swirl-like pattern. (**c**) New sensor working model wherein PNA is modified in a one-dimensional swirl-like pattern. Prior to hybridization, negatively charged markers can access and transfer electrons to the WE. After hybridization, the negative charge of the PNA/DNA duplexes imparts a negative potential to the WE through the π–π stacks and hinders the redox reaction of the marker over the entire WE.

**Figure 3 sensors-20-00836-f003:**
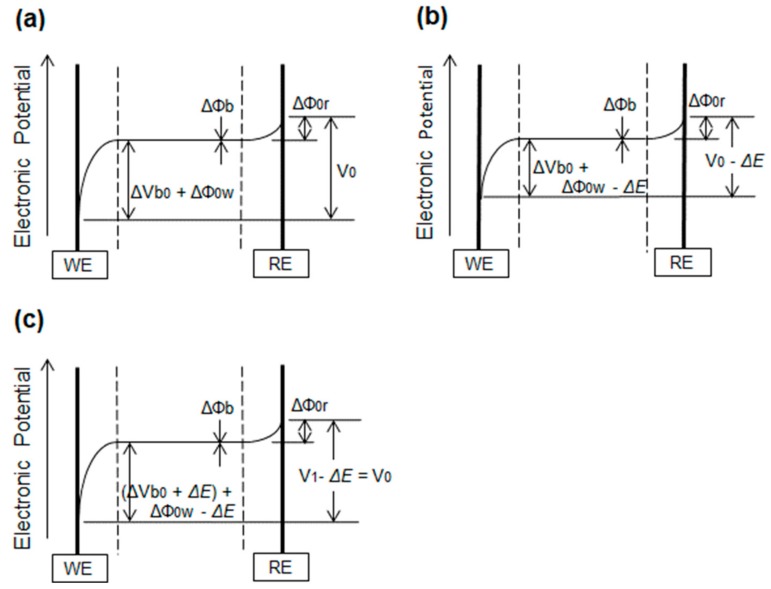
(**a**) Electric potential between the WE and reference electrode (RE) prehybridization for a measurement current (*im*) at an external bias of *V_0_*. (**b**) After hybridization, the electronic potential between the RE and WE decreases by *ΔE* due to the negative charge on the hybridized oligonucleotide, *ΔQ.* There will be a reduction in the current if the external bias is identical to *V_0_*. (**c**) To retrieve the identical current (*im*), the external bias *V_1_* must be increased by *ΔE*: (*V_1_* = *V_0_* + *ΔE)*.

**Figure 4 sensors-20-00836-f004:**
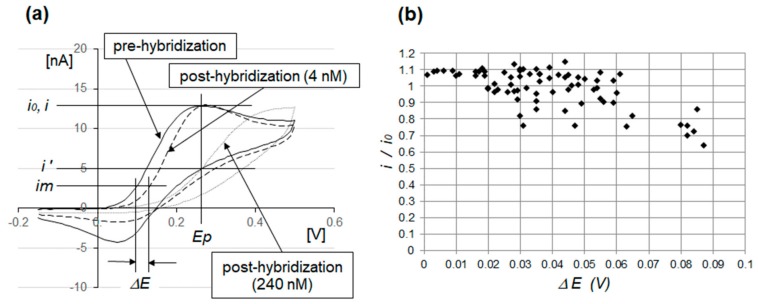
(**a**) Cyclic voltammograms (CVs) of gold electrodes modified with a PNA probe (N3) and 6-hydroxy-1-hexanthiol (6-HHT). The solid line indicates the prehybridization curve, while the dotted lines indicate the posthybridization curves at 240 nM and 4 nM of miRNA (R3). At 4 nM, the sensitivity of *i/i_0_* was poor, while the sensitivity of *ΔE* was high. (**b**) A 74-pc correlation chart for *i/i_0_* and *ΔE* is shown, representing when the complementary target miRNA (R3) concentration was 4 nM. Compared to the conventional *i/i_0_* evaluation parameter, *ΔE* has higher sensitivity and is more suitable for detecting low miRNA concentrations.

**Figure 5 sensors-20-00836-f005:**
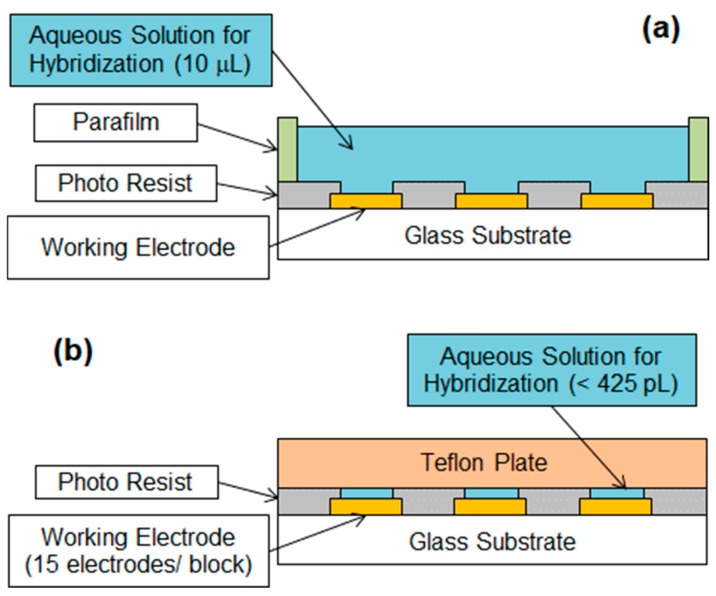
(**a**) Ordinary hybridization system and (**b**) new hybridization system, where a small well formed by an opening in the epoxy resin and Teflon plate limits the volume of the aqueous solution. The Teflon plate allows for the suppression of aqueous solution evaporation during hybridization and prevents the aqueous solution from being removed from each well when pressed to the array chip by a metal weight.

**Figure 6 sensors-20-00836-f006:**
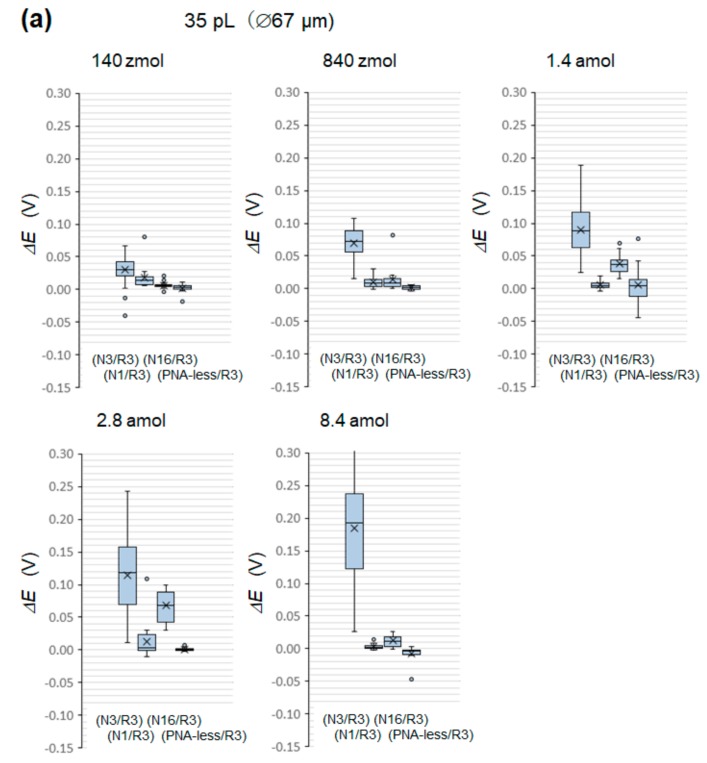
Boxplots of *ΔE* detection sensitivity data for (**a**) 35 pL (∅67 μm) and (**b**) 71 pL (∅95 μm) at varying molar amounts. In each box plot, (N3/R3) indicates 75 pcs of data of a complementary system, (N1/R3) and (N16/R3) each refer to 15 pcs of data of noncomplementary systems, and (PNA-less/R3) indicates 15 pcs of data of the reference system (to identify the occurrence of the chemisorption of miRNA onto 6-HHT).

**Figure 7 sensors-20-00836-f007:**
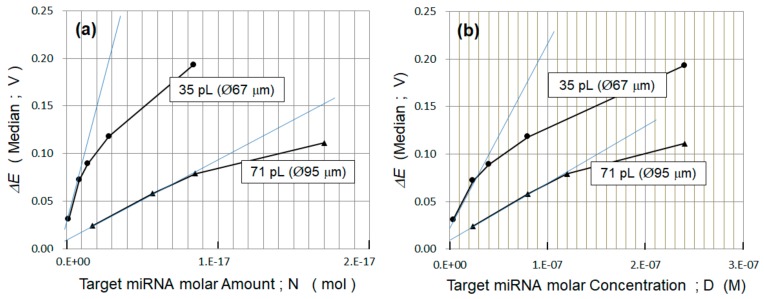
(**a**) The median values of the complementary system (N3/R3) in Figure 6a,b are plotted against the molar amount, *N*, on the linear scale, which is then (**b**) replotted against the molar concentration, *D*, on the linear scale. The highest *ΔE* detection sensitivity for the molar amount and concentration was 140 zmol and 4 nM, respectively, for the 35-pL well.

**Figure 8 sensors-20-00836-f008:**
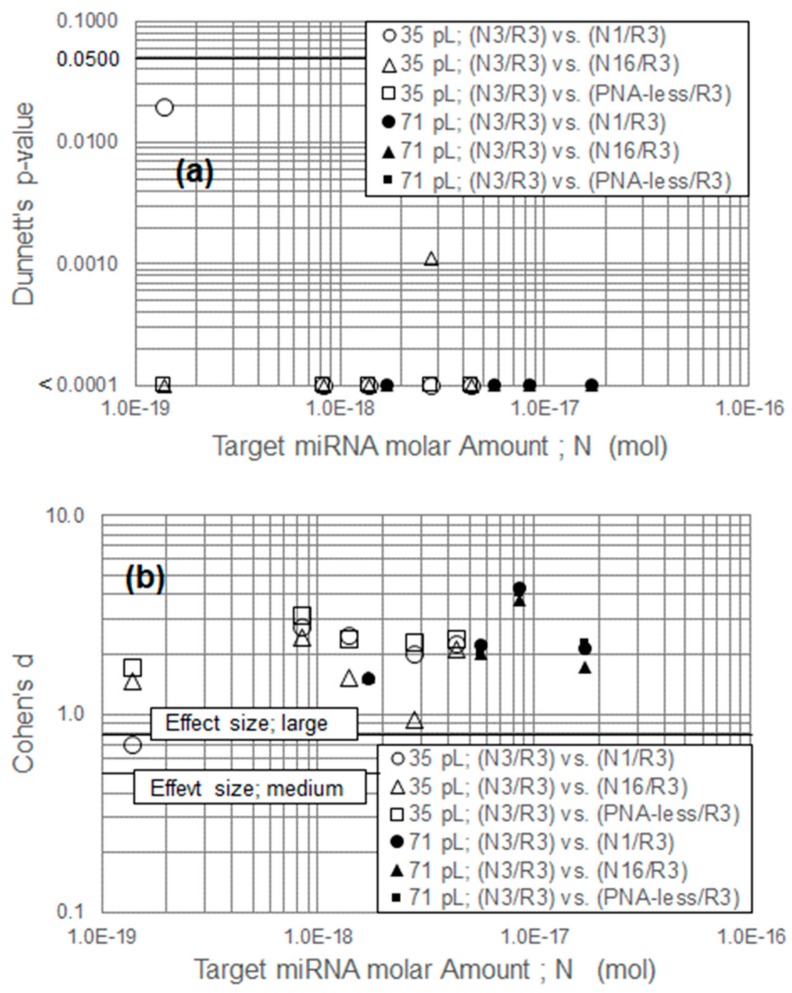
(**a**) Dunnett’s *p*-value and (**b**) Cohen’s *d*-value shown between the complementary and noncomplementary systems as (N3/R3) versus (N1/R3) and (N3/R3) versus (N16/R3). Computations were also performed between the complementary and PNA-less systems as (N3/R3) versus (PNA-less/R3). All computations were performed for the 35- and 71-pL hybridizations.

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
