# Peer review of "Label-Free Detection of Zeptomol miRNA via Peptide Nucleic Acid Hybridization Using Novel Cyclic Voltammetry Method"

_sensors, 2020, doi:10.3390/s20030836_

Round 1

Reviewer 1 Report

The authors developed a method to detect miRNA using cyclic voltammograms (CVs) and microelectrode array chips. The design is similar to a previous work but significantly improved. The results look sound and promising. I have some minor concerns that require the authors' attention. 

Figure 1a, there is no scale bar. Also, it is not clear which isi which for general readers. Figure b1 and b2 read confusing.  In Figure 2, is it better to have a slightly different diagram for the two measurements.  If the results heavily rely on the electrodes, how reproducible are the results obtained by different electrodes?  There are some typos in the text. 

Reviewer 2 Report

This work describes investigation of an electrochemical PNA detection method for miRNA, based on the approach in reference 6. The strengths of the work are some interesting conclusions regarding sensing in nanolitre droplets and a low sensitivity. The weaknesses of the work are the large errors on all measurements and the lack of sufficient comparison to literature. I recommend publication after fixing the following errors.

The authors should compare to alternate methods for miRNA sensing, for example electrochemical crispr. If the aim is to be highly sensitive to trace concentrations, cyclic voltammetry is a low sensitivity electrochemical technique (compared to pulsed or impedance methods) Does the 10 uL solution evaporate within the 30 mins at 25 C? If so then this pattern is likely due to the coffee ring effect seen in dropcasting and drying deposition and not due to electrostatics. Extra images of electrodes should be supplied in the supporting information, especially for different sized electrodes to test the change is the swirl pattern. The text on all figures is much too small The error bars on the data with the complimentary probes in Figure 5 are very large (probably due to using such small volumes with low concentrations) such that you would have no confidence of target levels from a measurement of delta E. Furthermore, the large variance in result for N1, N16 and PNA-free seem to suggest a large variance between electrodes which also limits the technology. 12 references are insufficient for a full paper. Please provide a better reflection of the work in literature and place the results in context.

Reviewer 3 Report

Manuscript ID: Sensors-683048

Title: Label-free Detection of Zeptomol miRNA via Peptide Nucleic Acid Hybridization Using Novel Cyclic Voltammetry Method

Authors: Shintaro Takase *, Kouta Miyagawa, Hisafumi Ikeda

Submitted to section: Biosensor

The authors have  evaluate the detection sensitivity of miRNA hybridization using PNA probes in cyclic voltammograms (CVs). We investigated the PNA probe modification pattern on an array  chip surface using fluorescently labelled cDNA and observed a one-dimensional swirl-like pattern.

They have also established a new model of the ion-channel sensor and introduced a novel evaluation parameter for hybridization. They have also predicted that  that the  sensitivity of ΔE will improve if the working electrode (WE) size is reduced. Before taking account into consideration of present form of manuscript for publication, authors need to modify manuscript for Sensors standards.

Comments:

In results and discussion sections, several parts looks like introduction which either removed or modify. Why there is no images of WE and other parts using SEM? Which can give the electrode picture before and after hybridization?Can authors perform IR of the WE to see the functional moities? If the PNA is  immobilized in Swirl pattern on the WE, then how the potential is uniform? Need explanation. Line 290-294……what that statement refer to? It appeared suddenly in the explanation. How PNA is immobilize on gold surface? Covalent conjugation? Or with SH group activation chemistry? Please check the spelling throughout the manuscript and the explanation of figures need to modify according to pictures. Section 2.2 and 2.3 Fabrication and array design are developed in house or taken from some article? Please provide the reference. Figure 5 need to enlarge as the contents in the figure is not visible Did authors validate the system and also compare with other techniques? What is the criteria for selecting 35 pL and 71 pL? Why not below and above that ?

Round 2

Reviewer 3 Report

The authors have addressed all the raised questions related with the scientific merits of the MS. Authors have further plans to submit another MS in future for some other supporting studies. My comments 4 and 11 are explained based on the authors further work on these aspects, which is absoutely absolutely understandable.